# Development of a Diabetes Dietary Quality Index: Reproducibility and Associations with Measures of Insulin Resistance, Beta Cell Function, and Hyperglycemia

**DOI:** 10.3390/nu16203512

**Published:** 2024-10-16

**Authors:** Maartje Zelis, Annemarie M. C. Simonis, Rob M. van Dam, Dorret I. Boomsma, Linde van Lee, Mark H. H. Kramer, Erik H. Serné, Daniel H. van Raalte, Andrea Mari, Eco J. C. de Geus, Elisabeth M. W. Eekhoff

**Affiliations:** 1Department of Internal Medicine, Amsterdam University Medical Center, VU University Medical Center, 1081 HV Amsterdam, The Netherlands; m.zelis@amsterdamumc.nl (M.Z.); amc.bik@gmail.com (A.M.C.S.);; 2Departments of Exercise and Nutrition Sciences and Epidemiology, Milken Institute School of Public Health, George Washington University, Washington, DC 20052, USA; 3Complex Trait Genetics, Center for Neurogenomics and Cognitive Research, Vrije Universiteit Amsterdam, 1081 HV Amsterdam, The Netherlands; 4Department of Human Nutrition, Wageningen University & Research, 6708 BP Wageningen, The Netherlands; 5Department of Internal, Vascular Medicine and Diabetes, Amsterdam University Medical Center, VU University Medical Center, 1081 HV Amsterdam, The Netherlands; 6Department of Internal Medicine, Endocrinology & Metabolism, Amsterdam University Medical Center, VU University Medical Center, 1081 HV Amsterdam, The Netherlands; 7CNR Neuroscience Institute, 35127 Padua, Italy; 8Department of Biological Psychology, Vrije Universiteit Amsterdam, 1081 BT Amsterdam, The Netherlands; eco.de.geus@vu.nl

**Keywords:** diet records, nutrition assessment, diabetes mellitus, beta cell, glucose metabolism, insulin metabolism

## Abstract

Aims: Various dietary risk factors for type 2 diabetes have been identified. A short assessment of dietary patterns related to the risk for type 2 diabetes mellitus may be relevant in clinical practice given the largely preventable nature of the disease. The aim of this study was to investigate the reproducibility of a short food frequency questionnaire based on available knowledge of diabetes-related healthy diets. In addition, we aimed to investigate whether a Diabetes Dietary Quality Index based on this questionnaire was related to metabolic risk factors, including measures of beta cell function and insulin sensitivity. Methods: A short food frequency questionnaire was composed by selecting fourteen questions (representing eight dietary factors) from existing food frequency questionnaires on the basis of their reported relationship with diabetes risk. Healthy participants (N = 176) from a Dutch family study completed the questionnaire and a subgroup (N = 123) completed the questionnaire twice. Reproducible items from the short questionnaire were combined into an index. The association between the Diabetes Dietary Quality index and metabolic risk factors was investigated using multiple linear regression analysis. Measures of beta cell function and insulin sensitivity were derived from a mixed meal test and an euglycemic–hyperinsulinemic and modified hyperglycemic clamp test. Results: Our results show that this new short food frequency questionnaire is reliable (Intraclass Correlations ranged between 0.5 and 0.9). A higher Diabetes Dietary Quality index score was associated with lower 2 h post-meal glucose (β −0.02, SE 0.006, *p* < 0.05), HbA1c (β −0.07, SE 0.02, *p* < 0.05), total cholesterol, (β −0.02, SE 0.07, *p* < 0.05), LDL cholesterol, (β −0.19, SE 0.07, *p* < 0.05), fasting (β −0.4, SE 0.16, *p* < 0.05) and post-load insulin, (β −3.9, SE 1.40, *p* < 0.05) concentrations and the incremental AUC of glucose during MMT (β −1.9, SE 0.97, *p* < 0.05). The scores obtained for the oral glucose insulin sensitivity-derived mixed meal test were higher in subjects who scored higher on the Diabetes Dietary Quality index (β 0.89, 0.39, *p* < 0.05). In contrast, we found no significant associations between the Diabetes Dietary Quality index and clamp measures of beta cell function. Conclusions: We identified a questionnaire-derived Diabetes Dietary Quality index that was reproducible and inversely associated with a number of type 2 diabetes mellitus and metabolic risk factors, like 2 h post-meal glucose, Hba1c and LDL, and total cholesterol. Once relative validity has been established, the Diabetes Dietary Quality index could be used by health care professionals to identify individuals with diets adversely related to development of type 2 diabetes.

## 1. Introduction

The pathogenesis of type 2 diabetes mellitus is complex, not fully understood, and influenced by both genetic and environmental factors. There is convincing evidence that intensive lifestyle interventions, including a healthy diet, can prevent the onset of type 2 diabetes among high-risk individuals, with lasting effects in the post-intervention period [1,2,3]. Evidence from mainly observational studies shows that diets rich in whole grains, fruits, vegetables, fish, poultry, and legumes; moderate in alcohol consumption; and low in refined grains, red or processed meats, and sugar-sweetened beverages are associated with a lower risk of type 2 diabetes [4]. For individuals at high risk of developing type 2 diabetes, assessment of such dietary patterns related to an increased risk for type 2 diabetes may contribute to better identification of those who need to adopt a healthier diet.

Most food frequency questionnaires (FFQs) are designed for research purposes to quantitatively assess intake of all foods consumed to enable quantification of macro- and micronutrient, and energy intake and evaluate their relationship with multiple health outcomes. These questionnaires are time-consuming to complete and analyze and are not intended for use in clinical practice. For health care practice, a short, reproducible and easily applicable FFQ could greatly enhance the feasibility of dietary assessment and aid health care professionals in developing individualized action plans to improve patients’ diets.

A number of brief dietary questionnaires have been developed and validated for assessing dietary patterns or specific food or nutrient groups [5]. However, most of them are not specifically designed to measure diets related to diabetes risk. One recent questionnaire was developed in the UK that assesses dietary habits in people at risk of diabetes [6]. So far, these short questionnaires have not been studied in relation to biological risk factors for type 2 diabetes.

The aim of this study was to determine the reproducibility of a short FFQ to assess dietary factors relevant to type 2 diabetes risk in healthy Dutch individuals. In addition, we investigated whether the Diabetes Dietary Quality index (DDQ-index) retrieved from this questionnaire was associated with type 2 diabetes-related metabolic risk factors, including measurements of beta cell function and insulin sensitivity (mixed meal- and clamp-derived measurements).

## 2. Materials and Methods

### 2.1. Questionnaire

A literature search was performed in the MEDLINE and Up to Date databases to identify studies on food consumption and the risk of type 2 diabetes. Reference lists of retrieved articles and topics were scanned for further studies and mostly observational studies were included.

Using the available knowledge on type 2 diabetes-related healthy diets and foods (components) [7,8,9,10,11,12,13,14,15,16,17,18,19,20], a food questionnaire was created). The questionnaire contains 14 questions on 10 items, including coffee, tea, alcohol, vegetable, fruit, cereal products, total fish, fatty fish, meat, and snack intake. Questions on snacks were added according to the emerging interest at that moment. The phrasing of the questions in the questionnaire, except those for fish intake, was based on a validated Dutch FFQ from the National Institute for Public Health and Environment and the PrimeScreen Questionnaire [21,22]. Questions about fish intake were obtained from a validated Dutch FFQ from Wageningen University [23]. Eight items from the questionnaire were selected to compose a Diabetes Dietary Quality Index.

### 2.2. Intake Assessment

Vegetable, fruit, fish, and meat intake was calculated in grams per day. Cereal fiber intake was calculated from bread, muesli/cruesli, cornflakes, Dutch rusk, and cracker intake. The Dutch national food consumption survey from 2003 was used to estimate the proportion of food subgroups if these were not specified. The mean fiber (from bread and cereal products) and alcohol content for each product was based on the Dutch NEVO-table [24].

### 2.3. Population

This study was part of a Dutch twin/family study that was conducted to determine the hereditary and environmental factors in the pathogenesis of type 2 diabetes mellitus among Dutch mono- and dizygotic twins and their siblings, as described in detail in previous studies [25,26]. The inclusion criteria were informed consent, European ancestry, a fasting plasma glucose < 7.0 mmol/l and age between 20 and 55 years. Subjects with metabolic disorders, Hemoglobin < 7.8 mmol, use of antivirals, corticosteroids, antihypertensive drugs and drugs that affect insulin secretion or insulin sensitivity and subjects with diabetes, serious heart, pulmonary, hepatic or renal diseases/impairment were excluded. Between September 2004 and the end of 2006, 190 subjects were included in this study and 176 (75 male, 101 female) completed the questionnaire. Without being alerted in advance, three months later, all 176 were asked to complete the questionnaire for the second time, of whom 123 subjects completed the questionnaire, with an intervening period of 3.15 months (SD 3.08 months).

### 2.4. Screening Visit

During the screening visit, a 75 g oral glucose tolerance test (OGTT) was performed to obtain fasting and 2 h glucose levels via a finger-prick [26]. The OGTT was performed as recommended by the World Health Organisation [27]. Blood glucose levels were determined with the HemoCue 201+ (Hemocue AB, Ängelholm, Sweden).

#### 2.4.1. Mixed Meal Test

After an overnight fast, subjects underwent a Mixed Meal test (MMT). Before the commencement of the MMT, a physical examination was performed, including weight, length and blood pressure measurements. The design of this test was described in detail in previous work [26].

#### 2.4.2. Clamp

At another visit, after a 12 h fast, the clamp procedure started in the clinic. The euglycemic–hyperinsulinemic and modified hyperglycemic clamp test design were described in detail in previous work [26].

#### 2.4.3. Laboratory Analysis

Blood samples derived during the mixed meal and clamp tests were determined for blood glucose bedside with a Yellow Springs Glucose Analyser. Blood for hormonal levels was centrifuged (1469× *g*) and the serum stored at −80 °C. All serum specimens were assessed to determine their insulin and C-peptide levels at the VU University Medical Centre (Department of Clinical Chemistry, Amsterdam, The Netherlands) using an immunometric assay luminescence method (ACS: Centaur; Bayer Diagnostics, Mijdrecht, The Netherlands).

### 2.5. Insulin Sensitivity and Beta Cell Function Parameters During MMT

Fasting and 2 h insulin levels and the insulin incremental area under the curve (AUC by the trapezium rule minus the fasting level) during the entire 4 h test were measured. To estimate the early insulin response, the insulinogenic index (insulin level t30 − t0/glucose t30 − t0) was calculated. Glucose levels were analyzed as follows: (1) the glucose level at 120 min and (2) the glucose incremental AUC during the period from 0 to 120 min and during the entire 4 h test. Overall glucose stimulated insulin secretion was calculated as the AUCinsulin/AUCglucose ratio and as the incremental AUC ratio (iAUCinsulin/iAUCglucose).

The homeostasis model assessment (HOMA) was used to assess insulin resistance (IR) and beta cell function using the fasting glucose and insulin concentrations: IR (fasting insulin in μU/mL × fasting glucose in mmol/L)/22.5 and beta cell function = (20 × fasting insulin)/(fasting glucose − 3.5).

Model-based beta cell function parameters were calculated using a mathematical model created by Mari et al. [28]. This calculation was also described in detail in a previous publication [25]. Three main outcome parameters were calculated: beta cell glucose sensitivity, potentiation factor ratio, and rate sensitivity.

As a measure of insulin sensitivity, oral glucose insulin sensitivity (OGIS) was estimated using the meal carbohydrate dose and glucose and insulin levels during the first 3 h of the meal test according to methods described by Mari et al. [29].

### 2.6. Insulin Sensitivity and Beta Cell Function Parameters During Clamp

The insulin sensitivity index (ISI) was defined as the glucose infusion rate per kg of body weight during the second hour of the euglycemic–hyperinsulinemic clamp per unit of serum insulin concentration (μmol kg^−1^ min^−1^ (pmol/L)^−1^). Figure 1 indicates the incremental insulin response and calculations related to the different secretagogues during a modified hyperglycemic clamp test.

### 2.7. Scoring

Dietary food items that showed good reproducibility were combined into the newly developed DDQ-index by using an existing scoring system developed by van Lee et al. in 2012 (The Dutch Healthy Diet Index) [30]. This is a continuous scoring system based on the Dutch Guidelines for a Healthy Diet. Coffee, tea and red meat were scored using the same method, but cut-off values were based on quantitative relationships between these food items and type 2 diabetes incidence in recent data from Dutch or European populations [31,32]. The scoring system and cut-off values are shown in Table 1. A higher score represents better adherence to a healthy diet and the maximum score per item is 10.

The scores for intake between threshold and cut-off value were allotted proportionally.

The DDQ-index was calculated as the sum of scores on separate food items; the total index has a range from 0 to 80 points.

### 2.8. Statistical Analysis

Mixed ANOVA with repeated measures as a fixed effect and family as a random effect were used to determine the reproducibility of the questionnaire. Differences were tested between the DDQ-index and constituents at time 1 and time 2.

To assess the test–retest reliability of the 10 variables and the agreement between the questionnaire at time 1 and time 2, we calculated the intraclass correlation (ICC) [33]. As the sample consists of individual (twin and sibs) nested families, we estimated the ICC by estimating the covariance matrix of each variable of the family members. All analyses were carried out in R; we used the OpenMx library to fit the model, as described, and to obtain the estimates of the ICCs and the associated 95%cIs.

Linear regression analysis was used to examine the association between the DDQ Index obtained from FFQ1 and cardiometabolic parameters and markers of insulin sensitivity and beta cell function. Regression models were run using the R gee package, with adjustment for age, sex, BMI, current smoking (yes/no), and current engagement in sports (yes/no). Family was used as a random effect to account for the clustering in the data.

## 3. Results

The FFQ was completed by 176 participants at baseline, and of these, 123 completed the questionnaire twice. Table 2 shows the subject characteristics.

Table 3 shows the mean intake determined from both questionnaires, and the test–retest reliability results of the comparison of the FFQ items at the two time points (intraclass correlation; ICC). The items showed no difference in mean intake and were highly correlated. For the 10 items scored, the ICCs ranged between 0.546 and 0.857.

The association between the DDQ index and the cardiometabolic parameters, insulin sensitivity, and beta cell function parameters retrieved during mixed meal and clamp tests are shown in Table 4 and Table 5. A higher DDQ-index score was inversely associated with 2 h post-meal glucose and HbA1c concentrations. The DDQ-index was also inversely associated with total cholesterol and LDL cholesterol concentrations (Table 4).

As shown in Table 5, the DDQ-index was significantly inversely associated with fasting insulin, 2 h post-meal insulin, the incremental AUC of glucose during the total 4 h mixed meal test, and the HOMA index for insulin resistance.

The DDQ-index was also significantly directly associated with insulin sensitivity during the first 3 h of the meal test (OGIS180). The DDQ-index was not associated with any clamp parameters.

## 4. Discussion

In this study, we designed a new short questionnaire assessing food groups related to the risk of type 2 diabetes and showed that this instrument was reproducible. ICCs were on the order or higher than standard accepted ICCs for FFQs (0.5 to 0.7) [34].

Based on this food questionnaire, we developed a new easily applicable DDQ-index, which was also based on the Dutch Healthy Diet index [30], and examined the relationship between physiological parameters, laboratory testing and non-physiological maximal testing of insulin production and insulin resistance. This index was associated with lower 2 h post-meal glucose, HbA1c, total cholesterol, and LDL cholesterol, fasting and post-load insulin, and the incremental AUC of glucose during MMT. Oral glucose insulin sensitivity (OGIS) derived from MMT was also higher in participants who scored higher on the DDQ-index.

Many dietary questionnaires exist, but most are elaborate and take a long time to fill out. Recently, the use of short food questionnaires has received more attention [5]. These questionnaires have shown good reproducibility, but they mainly focus on one aspect of the diet (e.g., fat intake [35,36,37,38]) or a limited number of food groups (e.g., only fruit and vegetables [39,40,41,42,43]) rather than different aspects of dietary patterns, or still include one day 25 item diary, as has been previously recommended for dietary management in pre-diabetic patients [6]. Only very few short one-time questionnaires have been related to a few specific chronic diseases or metabolic risk factors [44,45,46,47].

In 2013, van Lee at al. reported that the Dutch Healthy Diet (DHD)-index, which is based on 10 components derived from a 180-item FFQ, was directly associated with several micronutrients and concluded that the DHD-index could be used to estimate adherence to the Dutch dietary guidelines and as a monitoring tool in public health research [30,48]. The DHD index included information on physical activity, intake of vegetables, fruit, fish, and fiber, saturated fatty acids, trans fatty acids, frequency of consumption of acidic foods and beverages, sodium, and alcohol. They showed a good correlation of the FFQ-based DHD-index with a DHD-index based on two 24 h recalls [48]. However, they found no direct associations between the DHD-index score and cardiometabolic risk factors, despite an association with a lower risk of 20-year all-cause mortality in elderly people [48]. Nevertheless, the Healthy Eating Index, which is a comparable American composite tool that measures overall adherence to the Dietary Guidelines for Americans (DGA), has been linked to a lower risk of diabetes, cardiovascular diseases, and metabolic syndrome [49,50,51,52,53], and lower levels of cardiovascular risk factors such as LDL cholesterol [54]. Instead of these indexes based on elaborate FFQs, application of a much shorter and easier dietary assessment instrument could support easier assessment of dietary patterns in clinical practice and may serve as a practical tool to inform prevention or early treatment of various health conditions. This could, in particular, be relevant among a population at high risk of type 2 diabetes.

The DDQ questionnaire we developed consists of all items that emerged from a literature review on the relationship between diet and type 2 diabetes. To our knowledge, this questionnaire is one of the first short instruments to assess dietary factors related to diabetes risk.

In this study, a higher DDQ-index score was inversely associated with HbA1c, a risk factor and diagnostic criterion for type 2 diabetes (WHO2011). Recently, the Diabetes Prevention Program Research Group published a study which stated that that baseline HbA1c predicts diabetes incidence and can be reduced by lifestyle changes [55]. In our population, a difference of ten points (the maximum for one item) in the DDQ-index was associated with a 0.07 lower HbA1c in healthy individuals. A difference of ten points in the DDQ-index was associated with 0.23 mmol/l lower 2 h post-meal glucose levels. We did not find an inverse association between DDQ and 2 h plasma glucose during OGTT, which might be related to different mechanisms of glucose control due to the presence of proteins and fatty acids in the meal [56]. Furthermore, a higher DDQ-index was associated with better insulin sensitivity (as assessed by OGIS and IR HOMA), but not with measures of beta cell function retrieved from clamp testing or MMT modeling analysis. This may be explained by the fact that the clamp test consists of less physiologic beta cell potentiation stimuli, as it misses the enteral stimulation response to ingested glucose, like the incretin effect [57] and the first-pass extraction by the liver.

There is evidence that most of the incretin effect is due to the gut-derived incretin hormones glucagon-like peptide-1 (GLP-1) and glucose-dependent insulinotropic polypeptide (GIP), which are released during meal intake and stimulate insulin. GLP-1 secretion is reduced in type 2 diabetes patients after an oral glucose load and during a meal test, whereas stimulation of insulin secretion by GLP-1 infusion is relatively well preserved in diabetic patients. This may explain why the DDQ-index was associated with 2 h meal glucose levels but not with beta cell function retrieved from clamp testing [58].

After a meal, plasma glucose rises and the liver, which receives absorbed glucose through the portal vein, plays an important role in postprandial glucose homeostasis through production and uptake [59]. Failure of the liver to suppress its glucose production is an additional defect besides peripheral insulin resistance in diet-induced metabolic syndrome and type 2 diabetes and can occur before the development of peripheral insulin resistance [59]. The DDQ-index associated parameters are also determined by first-pass extraction by the liver.

Furthermore, a smaller subgroup participates in the clamp process, which could make it possible for these subgroups to differ in terms of metabolic control, insulin sensitivity, and other clinical characteristic [60].

A few limitations of this study should be considered. First, the short FFQ was developed in the Netherlands within a limited Dutch study population and the relative validity of this FFQ was not assessed in comparison with an extensive FFQ, dietary records, or biomarkers. However, the questionnaire items we used were taken from FFQs that have been shown to have reasonably good validity. This study does not provide information about lifestyle factors other than the issues mentioned. Although this does not hamper the applicability of the DDQ index in identifying individuals with unhealthy diets, more in-depth lifestyle assessment will be needed to tailor subsequent diet and lifestyle recommendations. Second, the food items for the questionnaire were selected in 2005 and 2006. Current evidence on, e.g., the relationship between sugar-sweetened beverages and type 2 diabetes is missing [61].

## 5. Conclusions

In conclusion, the currently developed short FFQ is reproducible and the derived DDQ-index was associated with several cardiometabolic risk factors in healthy individuals. The DDQ index was associated with measures of hyperglycemia such as HbA1c and with insulin sensitivity, but not with insulin secretion.

These results indicate that there may be a direct association between dietary factors previously associated with a lower risk of type 2 diabetes with HbA1C and insulin resistance as measured by physiological testing, while these links were not apparent from the non-physiological maximal beta cell stimulation tests and insulin sensitivity tests.

Since glucose spikes in themselves may play a toxic negative role in the genesis of type 2 diabetes [62], it can be hypothesized that this may contribute to the development of type 2 diabetes in the future.

Further studies and validation of the DDQ-index by investigating its relative validity against other dietary intake measures would be required before the index can be used to detect individuals who are eating diets that are adversely related to the development of diabetes. But once relative validity has been established, the simplicity of the DDQ-index could help health care professionals to identify individuals who are at risk.

## Figures and Tables

**Figure 1 nutrients-16-03512-f001:**
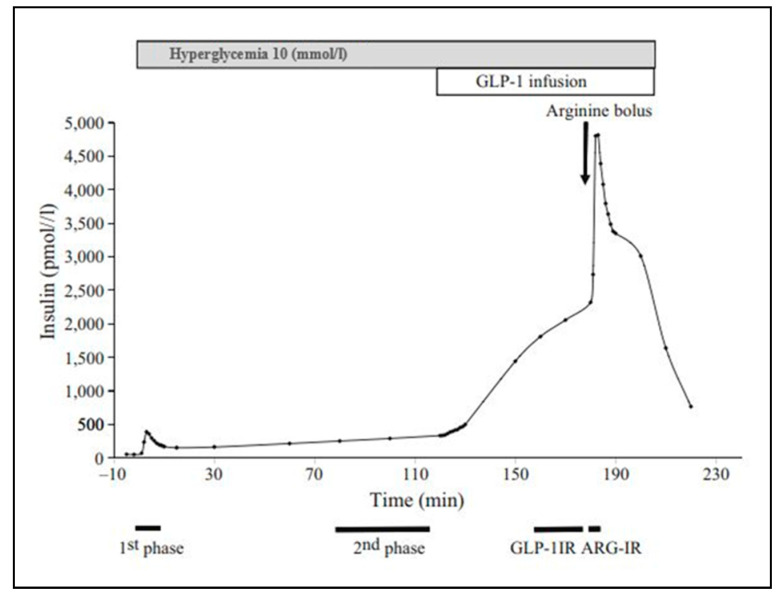
Insulin levels during the hyperglycemic clamp. The curve indicates the incremental insulin response to the different secretagogues.

**Table 1 nutrients-16-03512-t001:** The Diabetes Dietary Quality index (DDQ-index): rationale and calculation.

	Dutch Guidelines and Risk Factors	Minimum Score 0	Maximum Score 10
Vegetable (daily)	150 to 200 g of vegetables	0 g	≥200 g
Fruit (daily)	200 g of fruit (2 pieces)	0 g	≥200 g
Fiber (daily) (whole grains)	30 to 40 g a day of dietary fiber	0 g	≥40 g
Fish (daily)—omega-3-fatty-acids *	Two portions of fish a week, at least one of which should be oily fish. At least 450 mg omega-3-fatty-acids a day	0 mg	≥two portions oily fish or more frequent consumption of other fish
Alcohol (daily)	If alcohol is consumed at all, male intake should be limited to two Dutch standard units (20 g ethanol) a day and female intake to one.	Male: ≥60 g Female: ≥40 g	Male: ≤20 g Female: ≤10 g
Coffee (daily) **	Consumption of at least three cups per day may lower the risk of type 2 diabetes	0 cups	≥3 cups
Tea (daily) **	Consumption of at least three cups per day may lower the risk of type 2 diabetes	0 cups	≥3 cups
Red meat (daily) ***	There is a positive association between the consumption of red meat of a least 19 g a day and the incidence of type 2 diabetes	≥70 g	≤26 g

* Omega-3-fatty-acids intake from fish only. ** scores based on findings from van Dieren et al. [31]. *** scores based on findings from the InterAct Consortium [32].

**Table 2 nutrients-16-03512-t002:** Subject characteristics.

	All Subjects N = 176
Demographics	
Age (years), median (IQR)	31 (27–35)
Male (%)	42.6
Characteristics	
Body mass index (kg/m^2^), median (IQR)	23.3 (21.6–25.6)
Engaging in sports (%)	75
Current smoking (%)	32.4
Glucose metabolism variables	
Fasting glucose (mmol/L), median (IQR)	4.3 (4.1–4.6)
OGTT glucose at t120 (mmol/L), median (IQR)	5.4 (4.6–6.1) ^
Meal glucose at t120 (mmol/L), median (IQR)	5.3 (5.0–5.9)
IFG or IGT (%)	6.3 ^^
HbA1c (%), median (IQR)	5.3 (5.1–5.4)
Cardiovascular variables	
Total cholesterol (mmol/L), median (IQR)	4.2 (3.6–4.7)
LDL cholesterol (mmol/L), median (IQR)	2.2 (1.7–2.9)
HDL cholesterol (mmol/L), median (IQR)	1.42 (1.2–1.7)
Fasting triglycerides (mmol/L), median (IQR)	0.8 (0.6–1.0)
SBP (mmHg), median (IQR)	120.0 (112.5–128.3)
DBP (mmHg), median (IQR)	68.5 (112.5–128.3)
ALAT (U/l), median (IQR)	19.0 (15.0–27.0)
Diet	
Dutch Dietary Quality index score (max 80), median (IQR)	48.5 (42.2–54.4)

^ Values represent capillary whole-blood glucose levels (retrieved with Hemocue). ^^ IFG = Impaired fasting glycemia, IGT = Impaired Glucose Tolerance, according to the AMERICAN DIABETES ASSOCIATION classification. LDL = low-density lipoprotein, HDL = high-density lipoprotein, SBP = Systolic blood pressure, DBP = diastolic blood pressure, ALAT = alanine aminotransferase. HbA1c normal range = 4.3–6.1%.

**Table 3 nutrients-16-03512-t003:** Reproducibility of the 9 items retrieved from the short food-frequency questionnaire.

Foods and Food Components	Mean Intake (sd)	Mean Difference	*p* Value	ICC (95% cIs)
Coffee FFQ1 Coffee FFQ2 (cups (125 g) per day)	3.75 (3.96) 3.60 (3.45)	0.15	0.74	0.84 (0.77–0.89)
Tea FFQ1 Tea FFQ2 (cups (125 g) per day)	3.86 (3.80) 3.85 (3.97)	0.012	0.64	0.83 (0.75–0.88)
Alcohol FFQ1 Alcohol FFQ2 (g/w)	79.12 (80.15) 77.14 (91.27)	1.98	0.10	0.86 (0.75–0.91)
Vegetables FFQ1 Vegetables FFQ2 (g/d)	248.83 (102.20) 247.27 (95.46)	1.56	0.24	0.65 (0.48–0.77)
Fruit FFQ1 Fruit FFQ2 (g/d)	99.17 (70.96) 97.87 (71.25)	1.30	0.46	0.81 (0.74–0.87)
Fish FFQ1 Fish FFQ2 (Total amount) (g/w) Fish (fat) FFQ1 Fish (fat) FFQ2 (g/w)	71.06 (97.16) 84.60 (112.46) 21.18 (30.50) 23.63 (37.41)	−13.54 −2.45	0.96 0.39	0.55 (0.36–0.69) 0.75 (0.55–0.85)
Cereal fiber FFQ1 Cereal fiber FFQ2 (g/d)	12.15 (6.47) 12.62 (7.06)	−0.46	0.47	0.86 (0.79–0.90)
Red meat FFQ1 Red meat FFQ2 (g/d)	49.34 (31.55) 49,10 (28.43)	0.24	0.95	0.67 (0.54–0.77)
Snacks FFQ1 Snacks FFQ2 (frequency/d)	1.30 (0.85) 1.14 (0.77)	0.16	0.40	0.65 (0.51–0.76)

**Table 4 nutrients-16-03512-t004:** Linear regression analysis of the Diabetes Dietary Quality (DDQ)-index and cardiometabolic risk factors.

	DDQ-Index (10 Points Increment)
OUTCOME	
Fasting glucose (mmol/L)	0.02 (0.04)
OGTT glucose at t120 (mmol/L)	−0.06 (0.10)
HbA1c (%)	−0.07 (0.02) *
SBP (mmHg)	−0.64 (0.95)
DBP (mmHg)	−0.65 (0.73)
Total cholesterol (mmol/L)	−0.20 (0.07) *
HDL cholesterol (mmol/L)	0.02 (0.03)
LDL cholesterol (mmol/L) (calculated)	−0.19 (0.07) *
Fasting triglycerides (mmol/L) (ln-transformed)	−0.05 (0.04)
ALAT (ln-transformed)	0.09 (0.05)

Beta and SE values for a 10-point increment in the DDQ-index. All associations were adjusted for age, sex, BMI, current smoking (yes/no), and engagement in sports (yes/no). * significant at the 0.05 level.

**Table 5 nutrients-16-03512-t005:** Linear regression analysis (beta and SE) of the Diabetes Dietary Quality (DDQ)-index with beta-cell function and measures of insulin resistance during a mixed meal test (n = 176) and modified hyperglycemic clamp (n = 110).

Metabolic Parameters	DDQ-Index (10-Point Increment)
**Meal (classical)**	
Meal glucose at t120 (mmol/L)	**−0.02 (0.006) ***
Fasting insulin (pmol/L)	**−0.40 (0.16) ***
Serum insulin at t120 (pmol/L)	**−3.93 (1.40) ***
Serum insulin IAUC (0–240) (pmol·h/L)	**−4.06 (3.52)**
Glucose iAUC (0–240) (mmol·h/L)	**−1.91 (0.97) ***
Insulinogenic index	**0.00 (0.08)**
AUC_insulin_/AUC_glucose_ ratio (pmol/mmol)	**−0.21 (0.17)**
iAUC_insulin_/iAUC_glucose_ ratio (pmol/mmol)	**2.37 (2.08)**
Insulin resistance (IR HOMA)	**−0.01 (0.01) ***
Ln β-cell function (HOMA)	**−0.01 (0.00)**
**Meal (model-based)**	
β-cell glucose sensitivity (pmol min^−1^ m^−2^ [mmol/L]^−1^)	**0.14 (0.54)**
Rate sensitivity (pmol min^−1^ m^−2^ [mmol/L]^−1^)	**−0.86 (6.73)**
Potentiation factor ratio (220–240)/(0–20)	**<0.005**
Fasting ISR (pmol min^−1^ m^−2^)	**−0.17 (0.17)**
Integral of insulin secretion (nmol/m^2^)	**−0.05 (0.16)**
OGIS180 (mL min^−1^ m^−2^)	**0.89 (0.39) ***
**Clamp**	
Insulin sensitivity index (μmol min^−1^ kg^−1^ [pmol/L]^−1^)	**<0.005**
Insulin level at t0 of the clamp (pmol/L)	**−0.07 (0.24)**
iAUC of glucose-stimulated insulin from t1-t10 of clamp (pmol/L)	**3.85 (14.09)**
Second phase iAUC of glucose-stimulated insulin (pmol/L)	**−14.20 (77.89)**
GLP-1-stimulated iAUC of insulin (pmol/L)	**9.44 (282.11)**
Arginine-stimulated iAUC (pmol/L)	**12.93 (31.31)**

Analysis adjusted for age, sex, BMI, current smoking (yes/no), and engagement in sports (yes/no). iAUC = incremental area under the curve, ISR = insulin secretion rate. * significant at the 0.05 level.

## Data Availability

Data are contained within the article.

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
