# Peer review of "Development of a Diabetes Dietary Quality Index: Reproducibility and Associations with Measures of Insulin Resistance, Beta Cell Function, and Hyperglycemia"

_nutrients, 2024, doi:10.3390/nu16203512_

Round 1

Reviewer 1 Report

Comments and Suggestions for Authors

The topic - characterization of diet and risk of developing diabetes through reproducible and easy-to-apply instruments is a relevant topic

The summary is clear and appropriate

The introduction contextualizes and clarifies the objective of the study. The study question is identified

The methodology is described. The statistical methods are described. 

The results are presented in tables and graphs, and the text reflects the data considered most relevant

The discussion is appropriate and compares the results obtained with published works

The analysis of limitations is included in the discussion chapter , it does not mention that the study population is exclusively of European origin and applied in a single country using questionnaires developed for that country

The bibliography is current and relevant

The conclusions aligned with the results, 

questions:

line 108 Hb <7.8?

How did the authors conclude that all items have the same weight?

Author Response

Thank you very much for your valuable feedback and questions. This reviewer points out that our study population is exclusively of European origin and applied in a single country (the Netherlands) using questionnaires developed for that country. We agree with this point and revised the manuscript.

Questions:

1. Line 108: Hb < 7.8 mmol meaning hemoglobin. In the Netherlands we use mmol/l instead of g/dl. We have now written out the abbreviation Hb.

2. How did the authors conclude that all items have the same weight? We don't fully understand this question. I think it's about weighting the items within the DDQ index. The DDQ-index  is a continuous score.  For all components a maximum of ten points can be allotted, resulting in a range of zero to 80 points.  Cut-off values represent the required amount of consumption, as shown in table 1.

Reviewer 2 Report

Comments and Suggestions for Authors

General comments

In this the authors have evaluated the utility of a simple questionnaire of self-reported diet quality to evaluate the associated risk of insulin resistance, beta cell dysfunction and hyperglycemia with specific dietary factors.  This initial evaluation also tested the reliability of the index after test participants completed the questionnaire twice.  Overall, there was a high degree of reliability as indicated by the intra class correlations and there was a clear relationship between the estimated diet quality index and a suite of metabolic risk factors for T2DM.  The manuscript is well written, well organized and the results are convincing.  There are no major shortcomings, however, because of the limited sample size, as well as being confined to a single population of Dutch citizens enrolled in a twin/family study these data are very preliminary and this should be emphasized.  Follow up studies with a more heterogenous test population are needed.   

Author Response

Thank you very much for your valuable feedback. This reviewer points out that our study population has a limited sample size, as well as being confined to a single population of Dutch citizens enrolled in a twin/family study and therefore these data are very preliminary and this should be emphasized.  We agree with this point and revised the limitations.